# Obstructive Sleep Apnea and Adherence to Continuous Positive Airway Pressure (CPAP) Treatment: Let’s Talk about Partners!

**DOI:** 10.3390/healthcare10050943

**Published:** 2022-05-19

**Authors:** Debora Rosa, Carla Amigoni, Elisa Rimoldi, Paola Ripa, Antonella Ligorio, Miriam Fracchiolla, Carolina Lombardi, Gianfranco Parati, Elisa Perger

**Affiliations:** 1Istituto Auxologico Italiano, IRCCS, Department of Cardiovascular, Neural and Metabolic Sciences, 20149 Milan, Italy; 2Istituto Auxologico Italiano, IRCCS, SITR Lombardia, 20149 Milan, Italy; c.amigoni@auxologico.it; 3Ospedale San Giuseppe Gruppo Multimedica, Nursing Degree Course, University of Milan, 20122 Milan, Italy; elisa.rimoldi@multimedica.it (E.R.); paola.ripa@multimedica.it (P.R.); antonella.ligorio@multimedica.it (A.L.); 4Multimedica, IRCCS, Emergency Department, 20099 Sesto San Giovanni, Italy; miriam.fracchiolla@multimedica.it; 5Istituto Auxologico Italiano, IRCCS, Sleep Disorders Center & Department of Cardiovascular, Neural and Metabolic Sciences, San Luca Hospital, 20149 Milan, Italy; c.lombardi@auxologico.it (C.L.); gianfranco.parati@unimib.it (G.P.); e.perger@auxologico.it (E.P.); 6Department of Medicine and Surgery, University of Milano-Bicocca, 20126 Milan, Italy

**Keywords:** anxiety, depression, nursing, personal autonomy, quality of life, sleep

## Abstract

Background: Continuous positive airway pressure (CPAP) is the gold standard treatment for obstructive sleep apnea (OSA). The benefits of this therapy were studied and analyzed over time; patient adherence is often poor, as many factors negatively influence it. A topic that needs clarification is whether adherence to CPAP treatment in a patient with OSA is influenced by the behavior of a partner or spouse. Methods: A scoping review was conducted to evaluate the role of partner involvement in the CPAP treatment management in a patient with OSA. The research project was performed between August and September 2021 by consulting the main biomedical databases: CINHAL, Embase, PsycINFO, and PubMed. Results: Among 21 articles considered valid for our aim, 15 are qualitative studies, 5 are quantitative and 1 presents a mixed method. We identified several thematic areas and “key” elements, which are prevalent in the studies evaluated. Conclusions: The presence of a partner promotes adherence to CPAP therapy in patients with OSA, resulting in ameliorating their overall quality of life. To increase CPAP adherence, a trained nurse could represent a reference figure to technically and emotionally support couples during the adaptation period and in the long term.

## 1. Introduction

Obstructive sleep apnea (OSA) is characterized by frequent complete (apneas), or partial (hypopneas), obstruction of the upper airway (UA) during sleep [1]. It is estimated that 2 to 4% of normal-weight individuals, 3 to 28% of overweight individuals, and 40% of obese individuals suffer from this disease, with a higher prevalence in men compared to women [2,3]. A small and collapsible UA, and a lack of responsiveness in the pharyngeal muscles during sleep, are some of the key factors responsible for OSA [4,5]. Thus, structural alterations of the UA, such as adenotonsillar hypertrophy or craniofacial alterations, may induce UA collapse. Increased muscle collapsibility might also be determined by adipose tissue deposits around the neck in obese subjects, or hormonal and muscle changes during menopause [6,7,8] and in older individuals [8]. Comorbidities, excessive alcohol consumption, and medication-induced muscle relaxation, such as with benzodiazepines, might lead to UA restriction and collapse [4,9,10,11]. OSA is associated with major comorbidities, including daytime sleepiness in car accidents, impaired cognition, poor quality of life, hypertension, depression, fatigue [12], and sexual dysfunction [13]. Moreover, OSA worsens the prognosis of other conditions in patients with OSA, and is associated with increased overall, and especially cardiovascular, mortality [1,14,15,16].

In addition to the gold standard treatment, continuous positive airway pressure (CPAP), other treatment options include the use of mandibular advancement devices, surgery, weight loss, and positional therapy for certain specific phenotypes of patients with OSA [17]. In recent years, research moved towards alternative treatments for OSA, with the aim of precision medicine [18,19,20]. Recently, small, randomized, controlled trials (RCTs) on novel pharmacological therapies provided promising results [20,21,22].

Although several treatment options are available, CPAP often remains the first choice for treating OSA. Although CPAP contributes to the improvement of the cardiovascular, metabolic, and inflammatory parameters of the patient, and seems to reduce the risk of cardiovascular morbidity and mortality [23,24,25], up to 50% of patients with OSA refuse CPAP, or discontinue it within the first week [26,27]. To be effective, CPAP therapy should be used for at least 4 h per night [28], and up to 60% of patients who initially accepted CPAP treatment are not fully adherent [26,27,29,30]. Furthermore, in experienced patients who self-monitor using a mobile application, CPAP use is more consistent than in novice and intermediate users. In fact, experienced users have a significantly higher average use than both novice and intermediate users. Only 19.5% and 32.2% of novice and experienced patients, respectively, use CPAP for >20 min daily [31]. Among the reasons for reduced compliance with CPAP are CPAP-induced discomforts, such as mouth dryness, headaches, irritation, or ulceration of the oral and nasal skin [32]. Thus, recent literature focused on the best ways to improve CPAP compliance, such as involving sleep healthcare workers and telemedicine [33,34,35].

A survey conducted by the National Sleep Foundation found that 61% of adults sleep with a partner, and that one-quarter to one-third of married or cohabiting couples report that their intimate relationships are negatively affected by either excessive sleepiness, or problems with sleep [2,36]. A growing body of literature suggests that OSA is a shared problem that affects not only patients and caregivers but also partners; for example, long-term exposure to untreated OSA increases the risk of insomnia in partners [37]. Furthermore, partners may be distressed, not only by the presence of snoring, but also by hearing that the bed partner affected by OSA stops breathing during the night [38]. For these reasons, some couples decide to sleep in separate rooms, in order to improve the quality of their sleep [37]. Both OSA patients’ and partners’ sleep problems are associated with worse physical and mental health, well-being, social involvement, and quality of life [37]. It was established that partner involvement is important in the management of chronic diseases, particularly for therapeutic adherence, which seems to be influenced by the dyadic nature of the relationship between the patient and the partner [39,40,41,42,43].

Here, we present a review, with the aim of exploring the role of a partner in the process of treating OSA with CPAP.

## 2. Materials and Methods

We performed this scoping review according to the framework of Arkesey and O’Malley [43]. This framework consists of five steps: (1) identifying the research questions; (2) identifying relevant studies; (3) selecting studies; (4) extracting collected data; and (5) reporting results.

### 2.1. Identifying the Research Questions

To conduct this scoping review, the following research questions were identified:Is partner involvement effective in CPAP management of someone with OSA?What strategies are implemented by couples to improve quality of life?

### 2.2. Identifying Relevant Studies

Starting from the research questions, search terms were defined using the population, concept, and context (PCC) format [44]: OSA patients; CPAP therapy; and CPAP adherence of the couple. The following databases were consulted: Cumulative Index to Nursing and Allied Health Literature (CINAHL), Excerpa Medica dataBASE (Embase), psychological information database (PsycINFO), and PubMed. The following terms were used, introduced either as free terms or as Medical Subject Headings (MeSH) terms: “obstructive sleep apnoea syndrome”, “continuous positive airway pressure”, “treatment adherence and compliance”, and “spouses” (Table 1 search strategy). This was carried out independently by two reviewers, during the period between August and September 2021.

### 2.3. Selecting Relevant Studies

Studies were screened according to the PRISMA extension for scoping reviews (PRISMA-ScR) [45]. Two investigators completed the selection of studies after identifying and eliminating duplicate records using Mendeley. Inclusion criteria were studies describing the experiences of patients undergoing CPAP therapy for OSA, in English or Italian, patients aged 18 years or older, qualitative or quantitative primary studies, RCTs, observational studies, and published in the last ten years (2011–2021). Studies involving psychiatric and cognitively impaired patients were excluded.

### 2.4. Extracting Collected Data

All data were collected in a database. Data were extracted independently by two authors and classified into: author/year of publication; purpose; study design, sample, and methods; tools and strategies; results; and conclusion (Table 2).

### 2.5. Reporting Results

The results of this scoping review are summarized and reported by themes.

## 3. Results

Figure 1 shows the selection process of the studies, of which 21 met the inclusion criteria [61]. A total of 5 are qualitative studies [36,38,49,56,58], 15 are quantitative studies [23,28,29,46,47,48,51,52,53,54,55,57,59,60,62], and 1 has a mixed method approach [50].

The total number of people involved in quantitative studies is 2503 (range: 20–11,431); the total number of people involved in qualitative studies is 132 (range: 16–40), and the total number of people involved in the mixed method study is 24.

The following main focuses are identified: partner’s engagement, anxiety and depressive symptoms, daytime sleepiness and sleep quality, quality of sexual life and marital relationship empowerment, facilitators, and barriers to CPAP treatment.

### 3.1. Partner’s Engagement

Partner’s attitudes influencing patient’s device use can be classified as: (i) complete freedom (“patient autonomy”); (ii) shared management of the device (“supportive behaviour”); and (iii) complete substitution (“supervision and control”) [36]. “Supportive behaviour” consists of sharing CPAP management with the patient, by giving practice support and paying attention to the patient’s needs. This attitude also consists of helping with device starting, putting the mask on in the evening, adjusting the mask during the night when air leaks occur, and, generally, being present to manage problems [36,62].

Being supported and providing support allows patients and partners to become a “team.” Identification as a “we”, independent of CPAP use [58], increases the motivation for CPAP adherence [36,38,48]. A partner’s emotional support is associated with cooperation in treatment management within 3 months of beginning CPAP treatment [48]. This cooperation is particularly advantageous in reacting to a reduction in CPAP use [48]. Thus, in males and females married or cohabiting for more than one year, collaborative spouse involvement provides support and encouragement, and facilitates problem solving, with a positive impact on CPAP adherence [46,53]. In contrast, unmarried patients show greater ability and autonomy in managing the device by themselves [53]. In addition, the presence of other familiar components, such as children or grandchildren, appears to be crucial for CPAP adherence [23,38], and facilitates the establishment of proactive dynamics [53].

While educational and supportive interventions for both CPAP users and their partners during the first month of treatment are shown to increase adherence, this beneficial effect is not confirmed from 1 to 3 months, when only 1.6 h of CPAP compliance is reported [29]. After three years of CPAP use, spousal involvement seems not to be associated with treatment adherence, regardless of gender [55]. It is well known in the literature that adherence in the first month of therapy is a solid predictor of long-term use [30]. Thus, the positive involvement of the spouse at the beginning of CPAP use also determines long-term positive behaviour [30].

### 3.2. Anxiety and Depressive Symptoms

Increased energy levels and an improved quality of life, as consequences of OSA treatment, permit patients and their partners to re-engage in social and recreational activities [38]. Patients report that they do not feel tired during the day, and consequently, they feel happier and less irritable [38]. Moreover, the benefits observed over time from CPAP therapy increases patients’ desire to live longer [23].

### 3.3. Daytime Sleepiness and Sleep Quality

The Pittsburgh Sleep Quality Index (PSQI) [63], the Functional Outcomes of Sleep Questionnaire (FOSQ-10) [64], and the Epworth Sleepiness Scale (ESS) [65] are used to assess sleep quality, daytime sleepiness, and its impact on activities of daily living, respectively. Improvements in PSQI scores are reported in OSA patients after one month of CPAP treatment [60]. When education and support is provided to both the patient and their partner [29], the FOSQ-10 improves after 3 months of therapy, together with a decrease in daytime sleepiness [54]. The benefit of CPAP treatment on daytime sleepiness assessed with ESS at 3 months also results in a better relationship [29,60] and an overall improved quality of life [29,60].

### 3.4. Quality of Sexual Life and Marital Relationship

Among the consequences of OSA, sexual dysfunction has an emotional impact on patients, with significant effects on mood and social relationships [52]. In partners, sexual performance is described as unsatisfactory at the time of OSA diagnosis, with significant improvements after OSA treatment [52]. In male patients, OSA is typically associated with sexual dysfunction [51]. After 12 months of treatment, a beneficial effect is observed in terms of erection quality, ejaculation time, sexual desire [51], and improved sex life, more so in men than in women [52]. Furthermore, among males, marriage quality, as measured by the Dyadic Adjustment Scale (DAS) [23,46,54,55,59], is positively associated with CPAP adherence [55]. In contrast, in women, OSA often leads to psychological difficulties [47], and treatment of OSA for 12 months is associated with improvements in sex life quality [47,51]. In women with OSA, CPAP treatment does not adversely affect partner relationships [47].

Independent of gender, spousal involvement increases CPAP adherence after 6 months [55]. However, after 3 years of treatment, spousal involvement in the treatment group is no longer associated with adherence to therapy, even after stratification by gender [55].

### 3.5. Encouraging Empowerment

Empowerment and competence are identified as positive predictors of CPAP therapy, and are crucial for patients’ well-being and self-esteem [53]. Especially during the initial phase of CPAP treatment, the partner can help and support patient autonomy by improving spousal autonomy, sharing treatment management, and completing care of treatment management (“complete substitution”) [49].

An autonomy-promoting attitude in a partner increases the use of CPAP in OSA patients [53]. In fact, from data analyses from device use time, individuals with high autonomy support use the CPAP for an average of 166 min more than patients with low practical and emotional support [53].

### 3.6. Facilitators to CPAP Treatment

Encouraging partners with key attitudes, such as giving support in the practical management of therapy, using verbal encouragement for device use, and using humor to manage the situation, are considered relevant facilitators of increasing CPAP adherence [53]. A partner’s support and encouragement help the patient to feel less embarrassed, and consequently improves treatment adherence [36]. Positive factors for CPAP use include: (1) effective treatment and improvement of symptoms; (2) motivation; and (3) support from experts [49]. Couples believe that open, frequent, and supportive communication emphasizes and facilitates treatment adaptation [36].

Two studies report the involvement of experienced nurses [28,49]. In the first study [28], the nurse performs treatment education, explaining the fitting to all patients before the use of CPAP, follows up with the patients during the first week of treatment, and has follow-up visits at 3 months, 6 months, and then every 6 months [28]. In another study [49], experienced nurses provide information on OSA and adverse effects to both patients and partners. Couples report the need for support and assistance from healthcare professionals, which are perceived as an irreplaceable source of help and comfort [49].

### 3.7. Barriers to CPAP Treatment

Negative predictors of CPAP adherence include adverse effects, restriction of movement in bed, practical and psychosocial problems (e.g., difficulty in communicating with the mask), an inappropriate initial routine [49], feeling embarrassed or ‘unattractive’ in the eyes of the partner, and reduced adherence [36]. Furthermore, the ‘supervision and control’ of the partner is an attitude often judged as authoritarian, because of the imposition of directives that imply opposite reactions [36,62]. However, insistence on encouraging treatment, as well as derision, is not helpful for autonomy [53]. Popular culture and funny representations of snoring negatively impact adherence to CPAP treatment. Sometimes, men consider their wives’ concerns about their “not normal” snoring to be excessive. Conversely, some women report that their snoring is comically interpreted by their spouses, making the situation difficult and embarrassing, even to the point of not being able to discuss it seriously [36].

Negative feelings, such as ‘fear’ or ‘dread,’ explain why the diagnosis of OSA is often delayed [36]. Dying while sleeping due to prolonged apnea is often an erroneous fear that patients’ partners report and ask clinicians about during visits for sleep breathing disorders [36]. Anxiety is the predominant feeling at the beginning of treatment [53], which diminishes when the partner is supportive [35] and after CPAP treatment [36,60]. CPAP also seems to have a positive impact on depressive symptoms [36].

## 4. Discussion

The aim of this review was to assess the effectiveness of partner involvement in the CPAP management of OSA patients, and to analyze the strategies implemented by the couple to improve their quality of life. This scoping review identified the following themes: partner’s engagement, anxiety and depressive symptoms, quality of sexual life and marital relationship, encouraging empowerment, facilitators, and barriers to CPAP treatment.

Patients develop a ‘love-hate’ relationship with CPAP. In some interviews, patients often report that the first morning after treatment was the best day of their life [50]. Despite this, studies indicate variable adherence; thus, more than half (56%) of patients give up using the device within the first month [66]. Our review indicates that the presence of a companion who is emotionally supportive and who, through positive behaviour, encourages the patient to carry out therapy is fundamental in creating a bond of loyalty between the device and the patient. This is because, in patients with chronic diseases, it is essential that the therapy is carried out in a consistent manner [41].

Healthcare workers involved in sleep medicine also need to highlight the role of a supportive partner in motivating adherence at an early stage. Therefore, the sleep team should share information with patients and partners at the early stages of OSA diagnosis, in order to enable the beginning of more informed treatment, and to enable better adherence and greater motivation not to abandon it. Furthermore, it is important for healthcare professionals to assess whether a partner engages in pushy behaviors that compromise adherence [36].

Partner support should be maintained long-term to promote adaptation to the device, increased rest, and a reduction in anxiety and depressive symptoms, resulting in an improvement in the couple’s relationship and sexual satisfaction [47]. Recent studies show that the reduction of OSA episodes significantly improves daytime alertness [21], and both patients and their partners show an improvement in the couple’s quality of life.

From the analyzed studies, it emerges that an experienced nurse is considered by couples to be a strong point within the team [28,49]. Thus, the inclusion of a nursing figure could provide support to both the patient and the partner, as it is often not easy to maintain a high level of motivation. Indeed, after three years, the involvement of the spouse no longer seems associated with treatment adherence [55]. This may indicate the need for specialist nurses to provide long-term educational interventions. The nurse could act as a bridge between the sleep physicians and the couple, identifying difficulties and referring patients to different specialists, who could also help improve long-term motivation. This would be in line with the creation of personalized and supportive pathways for precision medicine in long-term interventions.

The literature still lacks knowledge on the link among treatment adherence, the patient–partner dyad, and the phenomenon of mutuality. While mutuality was investigated in several chronic pathologies [39,40,42], there is a need for studies on OSA patients with longer follow-ups. In addition, it would be useful to evaluate the effectiveness of various types of educational interventions, as well as motivational interviews carried out by experienced personnel. Furthermore, it would be useful to analyze data on adherence and mutuality within LGBTQ couples, to determine if there are differences compared to traditional partnerships. The authors of this review also propose mixed method studies investigating the role of the nurse as a facilitator of adherence, or as a substitute if the partner is not present or can take over this role.

### Study Limitations

The main limitations that were found in most studies are: the lack of sample assessment prior to CPAP treatment; a short time to determine long-term effectiveness on patients’ health; the use of different assessment tools but assessing the same outcomes (e.g., anxious-depressive symptoms, quality of life, and sleep–rest quality); small sample size; and no data were found on LGBTQ couples, which did not allow us to explain whether there are differences compared to traditional partnerships.

## 5. Conclusions

The findings of this review suggest that teamwork within the couple, allows the creation of a virtuous circle, and results in improved adherence to CPAP in OSA patients. Therefore, it is important for the sleep medicine team to be aware of the importance of the partner’s role in promoting CPAP adherence. An experienced nurse could act as a bridge between the sleep physicians and the couple [67,68], identifying difficulties and referring the patient to different specialists to improve long-term motivation [69]. This would pave the way for personalized and supportive pathways in OSA treatments.

## Figures and Tables

**Figure 1 healthcare-10-00943-f001:**
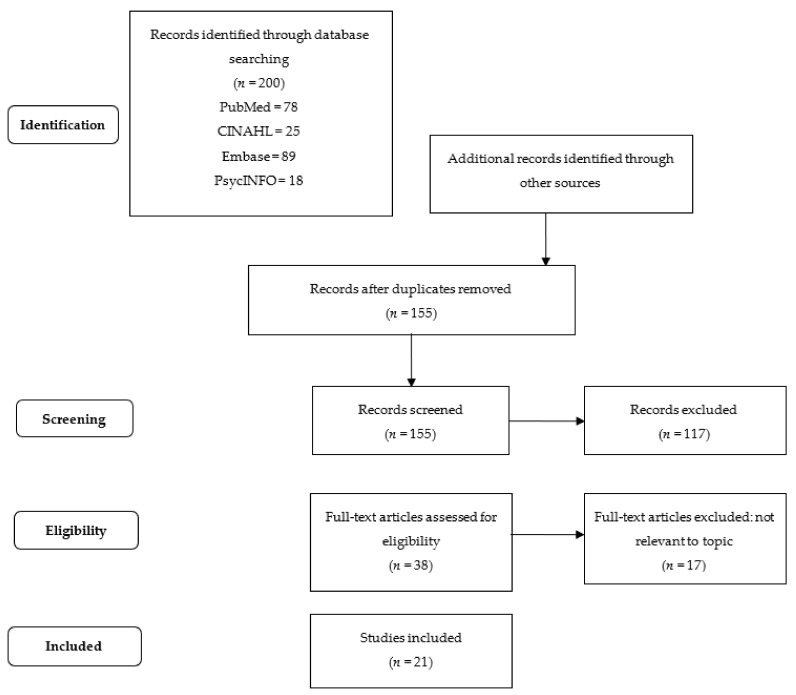
Workflow diagram of the search and selection process, based on the PRISMA flowchart.

**Table 1 healthcare-10-00943-t001:** Search strategy.

Database	Query
PubMed (1)	(“Sleep Apnea, Obstructive” [Mesh] OR “Sleep Apnea, Obstructive” [text word] OR Sleep Apnea, Obstructive OR “Apneas, Obstructive Sleep” [text word] OR “Obstructive Sleep Apneas” [text word] OR Sleep Apneas, Obstructive OR “Sleep Apneas, Obstructive” [text word] OR Sleep Apneas, Obstructive OR “Obstructive Sleep Apnea Syndrome” [text word] OR Obstructive Sleep Apnea Syndrome OR “Obstructive Sleep Apnea” [text word] OR Obstructive Sleep Apnea OR “OSAHS” [text word] OR OSAHS OR “Syndrome, Sleep Apnea, Obstructive” [text word] OR Syndrome, Sleep Apnea, Obstructive OR “Sleep Apnea Syndrome, Obstructive” [text word] OR Sleep Apnea Syndrome, Obstructive OR “Apnea, Obstructive Sleep” [text word] OR Apnea, Obstructive Sleep OR “Sleep Apnea Hypopnea Syndrome” [text word] OR Sleep Apnea Hypopnea Syndrome OR “Syndrome, Obstructive Sleep Apnea” [text word] OR Syndrome, Obstructive Sleep Apnea OR “Upper Airway Resistance Sleep Apnea Syndrome”[text word] OR Upper Airway Resistance Sleep Apnea Syndrome OR “Syndrome, Upper Airway Resistance, Sleep Apnea” [text word] OR Syndrome, Upper Airway Resistance, Sleep Apnea) AND (“Continuous Positive Airway Pressure”[Mesh] OR “Continuous Positive Airway Pressure”[text word] OR Continuous Positive Airway Pressure OR “CPAP Ventilation” [text word] OR CPAP Ventilation OR “Ventilation, CPAP” [text word] OR Ventilation, CPAP OR “Nasal Continuous Positive Airway Pressure” [text word] OR Nasal Continuous Positive Airway Pressure OR “nCPAP Ventilation” [text word] OR nCPAP Ventilation OR “Ventilation, nCPAP” [text word] OR Ventilation, nCPAP OR “Airway Pressure Release Ventilation” [text word] OR Airway Pressure Release Ventilation OR “APRV Ventilation Mode” [text word] OR APRV Ventilation Mode OR “APRV Ventilation Modes” [text word] OR APRV Ventilation Modes OR “Ventilation Mode, APRV” [text word] OR Ventilation Mode, APRV OR “Ventilation Modes, APRV” [text word] OR Ventilation Modes, APRV) AND (“Treatment Adherence and Compliance” [Mesh] OR “Treatment Adherence and Compliance” [text word] OR Treatment Adherence and Compliance OR “Therapeutic Adherence and Compliance” [text word] OR Therapeutic Adherence and Compliance OR “Treatment Adherence” [text word] OR Treatment Adherence OR “Adherence, Treatment” [text word] OR Adherence, Treatment OR “Therapeutic Adherence” [text word] OR Therapeutic Adherence OR “Adherence, Therapeutic” [text word] OR Adherence, Therapeutic) AND (“Spouses” [Mesh] OR “Spouses” [text word] OR Spouses OR “Spouse” [text word] OR Spouse OR “Married Persons” [text word] OR Married Persons OR “Married Person” [text word] OR Married Person OR “Person, Married” [text word] OR Person, Married OR “Persons, Married” [text word] OR Persons, Married OR “Husbands” [text word] OR Husbands OR “Husband” [text word] OR Husband OR “Domestic Partners” [text word] OR domestic partners OR “Domestic Partner” [text word] OR Domestic Partner OR “Partner, Domestic” [text word] OR Partner, Domestic OR “Partners, Domestic” [text word] OR Partners, Domestic OR “Spousal Notification” [text word] OR Spousal Notification OR “Notification, Spousal” [text word] OR Notification, Spousal OR “Wives” [text word] OR Wives OR “Wife” [text word] OR Wife OR “Spousal involvement” [text word] OR Spousal involvement)
PubMed (2)	(((caregiv* [TIAB] OR “CARE GIVER*” [Title/Abstract] OR SPOUS* [Title/Abstract] OR HUSBAND* [Title/Abstract] OR WIFE [Title/Abstract] OR WIVES [TIAB] OR MARITAL [TIAB] OR PARTNER* [Title/Abstract] OR COUPLE [TIAB] OR COUPLES [TIAB]) AND (CPAPS [TIAB] OR CPAP [Title/Abstract] OR “Continuous Positive Airway Pressure” [Title/Abstract] OR Ncpap [Title/Abstract] OR “Airway Pressure Release Ventilation” [Title/Abstract])) AND (“Obstructive Sleep Apnea*” [Title/Abstract] OR “Obstructive Sleep ApnOea*” [TIAB] OR OSAHS [Title/Abstract] OR “Obesity Hypoventilation Syndrome*” [TIAB] OR “Sleep Apnea Hypopnea Syndrome” [Title/Abstract] OR “Sleep Apnoea Hypopnea Syndrome” [TIAB] OR “Upper Airway Resistance Sleep Apnea Syndrome” [Title/Abstract] OR OSA [Title/Abstract] OR OSAS [Title/Abstract])) OR (((“Sleep Apnea, Obstructive” [Mesh]) AND (“Continuous Positive Airway Pressure” [Mesh])) AND ((“Caregivers” [Mesh]) OR “Spouses” [Mesh])) AND (y_10 [Filter])
CINAHL	((MH “Sleep Apnea, Obstructive”) OR TI (“Obstructive Sleep Apneas” OR “Sleep Apnoea Hypopnea Syndrome” OR “Obstructive Sleep ApnOea*” OR OSAHS OR “Sleep Apnea Hypopnea Syndrome” OR “Upper Airway Resistance Sleep Apnea Syndrome” OR OSA OR OSAS OR “Obesity Hypoventilation Syndrome*”) OR AB (“Sleep Apnoea Hypopnea Syndrome” OR “Obstructive Sleep ApnOea*” OR “Obstructive Sleep Apneas” OR OSAHS OR “Sleep Apnea Hypopnea Syndrome” OR “Upper Airway Resistance Sleep Apnea Syndrome” OR OSA OR OSAS OR “Obesity Hypoventilation Syndrome*”) AND (MH “Caregivers”) OR (MH “Spouses”) OR TI (caregiv* “CARE GIVER*” OR SPOUS* OR HUSBAND* OR WIFE OR WIVES OR PARTNER* OR MARITAL OR COUPLE OR COUPLES) OR (caregiv* “CARE GIVER*” OR SPOUS* OR HUSBAND* OR WIFE OR WIVES OR PARTNER* OR MARITAL OR COUPLE OR CUOPLES) AND (MH “Continuous Positive Airway Pressure”) OR TI (CPAP OR CPAPS OR “Continuous Positive Airway Pressure” OR Ncpap or “Airway Pressure Release Ventilation”) OR AB (CPAP OR CPAPS OR “Continuous Positive Airway Pressure” OR Ncpap or “Airway Pressure Release Ventilation”))
Embase	(((‘continuous positive airway pressure’/exp OR ‘cpap device’/exp) OR (cpap:ti,ab,kw OR cpaps:ti,ab,kw OR ‘continuous positive airway pressure’:ti,ab,kw OR ncpap:ti,ab,kw OR ‘airway pressure release ventilation’:ti,ab,kw)) AND (‘sleep disordered breathing’/exp OR (‘obstructive sleep apneas’:ti,ab,kw OR osahs:ti,ab,kw OR ‘sleep apnea hypopnea syndrome’:ti,ab,kw OR ‘upper airway resistance sleep apnea syndrome’:ti,ab,kw OR osa:ti,ab,kw OR osas:ti,ab,kw OR ‘obesity hypoventilation syndrome*’:ti,ab,kw OR ‘sleep apnoea hypopnea syndrome’:ti,ab,kw OR ‘obstructive sleep apnoea*’:ti,ab,kw)) AND ((‘caregiver’/exp OR ‘spouse’/exp) OR (‘caregiv* care giver*’:ti,ab,kw OR spous*:ti,ab,kw OR husband*:ti,ab,kw OR wife:ti,ab,kw OR wives:ti,ab,kw OR partner*:ti,ab,kw OR marital:ti,ab,kw OR couple:ti,ab,kw OR couples:ti,ab,kw)) AND (2011–2021)/py) AND (‘article’/it OR ‘article in press’/it OR ‘chapter’/it OR ‘conference paper’/it OR ‘review’/it)
PsycINFO	sleep apnea exp spouses/or exp “marriage and partner measures” exp Caregivers (“Obstructive Sleep Apnea*” or “Obstructive Sleep ApnOea*” or OSAHS or “Sleep Apnea Hypopnea Syndrome” or “Sleep Apnoea Hypopnea Syndrome” or “Upper Airway Resistance Sleep Apnea Syndrome” or OSA or OSAS or “Obesity Hypoventilation Syndrome*”) (“caregiv* CARE GIVER*” or SPOUS* or HUSBAND* or WIFE or WIVES or PARTNER* or MARITAL or COUPLE or COUPLES) 1 or 4 2 or 3 or 5 (CPAP or CPAPS or “Continuous Positive Airway Pressure” or Ncpap or “Airway Pressure Release Ventilation”) 6 and 7 and 8 limit 9 to yr = ”2011–Current”

**Table 2 healthcare-10-00943-t002:** Data extraction.

Author/ Year	Aim	Study Design, Sample	Tools and Strategies	Results	Conclusion
Baron et al., 2011 [46]	To determine the effects of spouse involvement on CPAP adherence and response to treatment problems in male patients with OSA.	Quantitative observational study. Sample 31 males. Inclusion criteria: age < 65 years; cohabitants > 1 year; new diagnosis of OSA; never used CPAP. Exclusion criteria: chronic obstructive pulmonary disease; oxygen therapy; congestive heart failure; cardiomyopathy; psychosis. Study length: 3 months.	Daily questionnaire to be completed by the patient in the evening before going to bed, without the help of the wife, to study the bidirectional relationship between spouse involvement and nocturnal CPAP adherence. Patients rate the wife’s behaviors on a scale: pressure to use the device, cooperation, and support. Severity of illness → AHI; quality of relationship → QRI	94% report emotional support from their spouse; 77% report being helped with CPAP management; 63% report that their partner reminded them to perform the treatment. The presence of emotional support is associated with cooperation in treatment management (*p* = 0.06). Spousal support increases after nights with reduced adherence to therapy (*p* < 0.05). Couples with a positive relationship have higher treatment adherence and cooperation (*p* < 0.05). Wife pressure to use CPAP is negatively correlated with objective adherence at 3 months.	Emotional and practical involvement of the spouse improves the patient’s approach and adherence to treatment. High levels of relational conflict may interfere with collaboration.
Gagnadoux et al., 2011 [28]	To assess the impact of socioeconomic factors on long-term treatment adherence, patient characteristics, and OSA prior to initiation of CPAP therapy.	Multicentre prospective cohort study. Samples: 1141 (674 CPAP adherent, 467 non adherent). Inclusion criteria: age > 18 years; CPAP treatment. Exclusion criteria: patients with mental retardation; patients unable to give informed consent; unable to read and/or speak French; with neuromuscular diseases. Study length: 6 months.	CPAP pre-treatment: health education by a specialist nurse. Tools: Subjective daytime sleepiness → ESS; depressive symptoms → Pichot scale, SES (self-administered questionnaires). One week later, telephone calls were made by the nurse. Follow-up: 3, 6 months. Daily use of CPAP, recorded via device memory card.	Non-adherence is associated with: working patients, non-cohabiting, normal weight, mild to moderate OSA, and smokers (*p* = 0.051). CPAP adherence is associated with four variables: AHI (*p* = 0.003); BMI (for BMI ≥ 25 and <30 kg/m², *p* = 0.01; for BMI ≥ 30 kg/m², *p* = 0.01); employment status (*p* = 0.007); married marital status (*p* = 0.01). Depressive symptoms and daytime sleepiness does not show a statistically significant difference in the non-adherent group (*p* = 0.18 for Pichot; *p* = 0.85 for ESS).	Adherence to CPAP influenced by: partner’s post-treatment sleep quality and quality of life. Patients who live alone and/or work are at higher risk of non-adherence.
Petersen et al., 2011 [47]	To investigate the effects after 1 year of CPAP treatment on difficulties, discomfort, and sexual dysfunction in female patients with OSA	Quantitative observational study. Sample: 92 female patients. Inclusion criteria: diagnosis of OSA; CPAP treatment; age > 18 years; ability to read and write in Danish. Exclusion criteria: sleep disorders; psychiatric comorbidity. Study length: 1 year.	The sample responded to mailed questionnaires and to the 1 year follow-up in CPAP treatment. Tools: sexual performance for women with partner → FSFI; sexual difficulties → FSDS; sexual difficulties → MFSD; life as a whole, family life, relationship with partner, and sexual life → LiSat-11; daytime sleepiness →ESS.	The FSFI results show no significant improvements for any of the items: desire (*p* = 0.69); arousal (*p* = 0.97); lubrication (*p* = 0.85); orgasm (*p* = 0.90); satisfaction (*p* = 0.96); pain (*p* = 0.94); total score (*p* = 0.89). FSDS results show no significant improvement (*p* = 0.06). MSFD results show a reduction in sexual difficulties in the older age group (≥45 years) (*p* = 0.06) compared to the younger (<45 years) (*p* = 0.63). The LiSat-11 results do not show significant improvements for any of the parameters: life as a whole (*p* = 0.59); family life (*p* = 0.73); relationship with partner (*p* = 1.00); sex life (*p* = 0.92). The results of the ESS scale show a significant improvement after 1 year of CPAP treatment (*p* < 0.001).	CPAP treatment does not adversely affect family or partner relationships. Sharing this information with patients can be important when starting treatment.
Baron et al., 2012 [48]	To assess spouse involvement in CPAP treatment of the person with OSA, and its association with adherence to therapy.	Longitudinal observational study. Sample: 23 male. Inclusion criteria: age 18–65 years; male gender; diagnosis of OSA; married or cohabiting with a partner ≥1 year; CPAP treatment. Exclusion criteria: CPAP use by spouse; chronic obstructive pulmonary disease; oxygen therapy; conditions such as: heart failure, cardiomyopathy, psychosis; use of other concomitant treatments for OSA. Length of treatment: 3 months.	Demographic questionnaire completed by patients, prior to CPAP treatment. Partner involvement assessed with questionnaires 7–10 days post treatment start. Tools: Assessing spousal support for CPAP support → Actions Scale-C32, 1 week after treatment start and after 3 months.	At 3 month follow-up, *N* = 14 Spousal involvement is rated positively by 83% of the patients, with a mean rating of 2.3. A total of 57%, with a mean rating of 1.9, also report negative behaviour. Spousal involvement increases at follow-up, although not significantly (*p* = 0.07). Adherence to therapy at 3 months improves, and is statistically significantly (*p* = 0.002).	Involvement of the spouse, especially if positive and supportive, in CPAP treatment in male patients with OSA increases adherence to therapy from the start of treatment to 3 months.
Elfström, et al., 2012 [49]	To explore and describe the factors that influence partner support in patients with OSA and how they manage these situations during the initial phase of CPAP treatment.	Exploratory qualitative study. Sample: 25 partners of OSA patients treated with CPAP (18 females and 7 males). Inclusion criteria: Age > 18 years. Exclusion criteria: Patient or partner with a life-threatening illness; diagnosis of a severe psychiatric illness; diagnosis of dementia; difficulty in reading or speaking Swedish.	Semi-structured interviews, lasting from 12 to 60 min, were administered to the partners. Interviews were based on three open-ended questions, asking for a description of: a situation that facilitated treatment support; a situation that worsened the support; the management of these situations.	Five negative factors emerge: adverse effects, limited effect, practical and psychosocial problems, and inappropriate initial routine. Four positive factors emerge: effective treatment, improvement, motivation, and support. Three behaviors influence the partners’ support in using the device: complete freedom (patient autonomy), shared management (supportive behaviour), and substitution (supervision and control).	The presence of the partner on the first days of treatment is a positive predictor of treatment adherence.
Henry & Rosenthal, 2013 [50]	To illustrate the significance of the role and relationship with the partner in OSA patients, to diagnose, manage, and set up treatment.	Mixed method. Samples: 24 (12 patients and 12 partners)	Twenty-four semi-structured, qualitative–quantitative interviews were conducted.	A total of 10 patients (83%) report that the symptoms of OSA were identified by their partners: “my husband used to wake me up and say: Hey, you’re not breathing”. Male snoring is often considered “normal” compared to female snoring. Snoring is often mocked by the spouse in contrast to apnea. Patients often associate daytime symptoms with the disease. For 50% of the sample, body weight is the main cause of the problem. The willingness to tackle the problem is delayed. Love–hate relationship with CPAP. Three partners complain about the noise or intrusiveness of the machine, even though they are in favor of the treatment; three spouses, on the other hand, report relaxing because they no longer have to control their partner’s breathing.	It is found that the role of the spouse is crucial in shaping problem identification, perception, help-seeking, and evaluation of the effectiveness of CPAP treatment in patients with OSA.
Petersen et al., 2013 [51]	To investigate the impact, after 1 year of CPAP treatment, on sexuality in male patients diagnosed with OSA.	Quantitative observational study. Sample: 146 patients. Inclusion criteria: diagnosis of OSA; CPAP therapy; age > 18 years; ability to read and write in Danish. Length of study: 1 year.	Tools: Life satisfaction → LiSat-11; sexual function → FSFI; daytime sleepiness → ESS.	The LiSat-11 questionnaire shows that sex life significantly improves after 1 year with CPAP treatment (*p* < 0.05), while family life and relationship with the partner does not change significantly (*p* > 0.05). All four items of the BSFI change in a statistically significant way: erection (*p* < 0.05); ejaculation (*p* < 0.01); desire (*p* < 0.001); evaluation of problems (*p* < 0.001). ESS improves significantly after 1 year of CPAP treatment (*p* < 0.0001).	Significant improvement in sex life and performance in male patients with OSA after 1 year of CPAP treatment.
Acar et al., 2016 [52]	Assessing sexual performance in partners before and after CPAP therapy in men with OSA.	Prospective study. Sample: 31 male patients. Patient inclusion criteria: Age > 18 years; AHI ≥ 25; CPAP therapy; BMI ≤ 40 kg/m²; normal uro-andrological examination. Patient exclusion criteria: nitrate treatment and erectile dysfunction (ED) treatment; non-heterosexual relationship; altered hormonal status; diagnosis of: hypertension, diabetes mellitus, peripheral neuropathic disease, or prostate cancer; renal transplantation; aortic aneurysm; spinal cord injury; endocrine disorders, and chronic and acute psychiatric disorders; penile deformities; alcohol abuse; psychotropic drugs; chronic diseases and cardiovascular diseases; and metabolic disorders and neurological disorders. Partner inclusion criteria: age > 18 years; normal urogynecological examination; Partner exclusion criteria: history of alcohol or other substance abuse; severe cardiac or pulmonary disease; uncontrolled hypertension, diabetes mellitus, thyroid disease; history of medication use with sexual side effects; severe pelvic organ prolapse. Length of study: 3 months.	Patients and their partners completed questionnaires separately before CPAP treatment and at 12 weeks, without sharing results. CPAP use was assessed through the device’s internal memory device. Tools: patient sexual function → IIEF; female partner sexual function → FSFI; depression → BDI.	Based on the IIEF questionnaire, all aspects of male sexual functioning improve significantly after CPAP therapy (*p* < 0.01). An improvement in FSFI post-treatment is also observed in partners (*p* < 0.001). Assessment of BDI in women at 12 weeks improves statistically significantly (*p* < 0.01).	CPAP therapy led to improvements in all aspects of male sexual performance. In partners, it was assessed that sexual performance may be unsatisfactory at the diagnosis of spouse OSA, but improves with CPAP treatment. In addition to improving quality of life in men with OSA receiving CPAP treatment, BDI results indicate that the emotional benefits of treatment also extend to the psychological state of the partner.
Luyster et al., 2016 [38]	To explore the experiences and difficulties of patients and their partners with CPAP use. Identify an introductory CPAP coaching programme.	Qualitative study design with focus group. Sample: 26 (14 patients and 11 partners, of which 3 male and 8 female). Inclusion criteria: age > 21 for both patients and partners; OSA patients undergoing CPAP therapy; cohabiting for at least 1 year.	Eight focus groups were conducted. Patients and partners participated separately in a total of four groups (3–4 participants per group). Each of the focus groups lasted about 90 min. Half of the focus groups were conducted in person and half by telephone	Five themes are identified: knowledge of sleep apnea; effects of sleep apnea; effects of CPAP; barriers and facilitators to CPAP; ideas for a new user support programme.	The inclusion of the partner in CPAP treatment is identified as a key component of treatment adherence. The partner is a facilitator for device use and identification of strategies in the early phase of therapy. Both patients and their partners experience the disease negatively. The study suggests that the ways in which couples interact in the face of chronic illness evolve as the different stages of chronic illness occur.
Baron et al., 2017 [53]	Assessing the factors promoting CPAP adherence in women, and the change in the quality of the relationship with the partner.	Pre-post post-test study. Sample: 20 women (13 married/cohabiting and 7 unmarried/non-cohabiting) Inclusion criteria: age between 18 and 70 years; future treatment with CPAP for OSA. Exclusion criteria: diagnosis of chronic obstructive pulmonary disease; neurological disorders; oxygen therapy; future surgery (3 months); use of other treatments for OSA; dementia; inability to read or write English; unstable psychiatric disorders. Study length: 3 months.	Pre-treatment questionnaires were administered to participants. Adherence to CPAP was assessed at 12 weeks using the device memory. Tools: relationship quality → QRI; emotional support → QRI. One week after the start of CPAP, seven participants provided answers to open-ended questions about the effective and non-effective behaviors with which their partner was involved in therapy.	Higher CPAP adherence among married/cohabiting participants (*p* < 0.08). Relationship conflict was negatively associated with treatment adherence (p < 0.05). Greater ability to self-manage treatment reported by the unmarried (*p* < 0.05). Most (6 out of 7) of the participants reported support and encouragement from their spouse/partner. Favourable spouse attitudes: support, encouragement, humorism. Non-favourable attitudes: insistence/harassment, taunting. Participants did not demonstrate changes in the quality of the relationship.	This study assesses that a supportive relationship is important for women’s use of CPAP. Understanding the factors that influence CPAP therapy reduces the risk of non-adherence.
Tramonti et al., 2017 [54]	Assessing the quality of relationships in a sample of patients with OSA treated, or not treated, with CPAP.	Quantitative observational study. Sample: 87 (71 males and 16 females; 28 treated with CPAP and 59 untreated) Exclusion criteria: significant comorbidities; unstable medical conditions; shift work; not married; cohabitation < 1 year. Study duration: 6 months.	Tools: subjective daytime sleepiness → ESS; relationship quality → DAS. Adherence to treatment was assessed through the device’s internal memory device.	Treated patients have lower AHI and ESS scores than untreated patients (*p* = 0.46). Untreated patients show lower DAS scores in the items: affective expression and spousal support (*p* = 0.046). There are no differences between men and women in either group. Age and relationship duration are positively correlated with total DAS scores in the CPAP-treated group (*p* ≤ 0.01).	OSA symptoms have an impact on marital satisfaction and relationship quality. The relationship quality of CPAP-treated patients is better than that of untreated patients. Adequate CPAP treatment is important, not only for the clinical condition, but also for the improvement of quality of life. Lasting relationships can act as resources for adaptation to CPAP.
Batool-Anwar et al., 2017 [55]	Determine whether spousal involvement affects adherence to CPAP therapy, and how this association varies with gender.	Multicentre randomized double-blind study. Samples: 194 (84 sham CPAP, 110 CPAP). Inclusion criteria: Age ≥ 18 years; clinical diagnosis of OSA; AHI ≥ 10. Exclusion criteria: previous treatment for OSA with CPAP or surgery; oxygen saturation on baseline PSG < 75% by >10%; history of motor vehicle accident related to sleepiness in the last 12 months; presence of chronic diseases; hypnoinductive drugs; shift work. Study length: 3 years.	Patients were administered the 32-item DAS questionnaire assessing marital/affective relationships. CPAP use was recorded on the device’s memory card and analyzed. Long-term CPAP adherence was measured as self-reported at the time of DAS administration. Follow up: 6 months and 3 years.	After randomization at 6 months, CPAP adherence and spouse involvement emerges only in the CPAP group (*p* = 0.01). After gender stratification, the association between spousal involvement and CPAP adherence is limited to men only, in a statistically significant manner (*p* = 0.03). Spousal relationship quality is not associated with treatment adherence. At three years, spousal involvement in the CPAP group is not associated with treatment adherence (*p* = 0.13), even after stratification by gender.	Spousal involvement is important in determining CPAP adherence during the initial treatment period, but has no effect on long-term adherence.
Ye et al., 2017 [36]	Identifying aspects that promote and influence CPAP success in conjugate OSA patients.	Exploratory descriptive qualitative design. Sample: 20 couples Inclusion criteria: Age ≥ 18 years; OSA patient; CPAP; married/cohabiting for at least 1 year. Exclusion criteria: not cohabiting; working at night; not understanding English; partners of eligible patients should not have had OSA or been treated with CPAP.	Semi-structured interview of seven questions. The pairs were interviewed together. The interviews focused on the couples’ learning about CPAP management. The interviews lasted approximately 40–60 min, and were conducted in the field.	Sleep disruption and the patient’s health are the reasons why the partners encouraged the start of treatment. It is important that there is complicity in the couple. Couples report that CPAP improves sleep quality and couple’s life. Partners are aware that they are often the reason why their spouse initiates treatment. Both verbal and practical support from the partner in using CPAP was important. Limitations reported for the use of CPAP are: disruption of bedtime routines; decreased intimacy; patients’ concern for their image.	The role of partners is crucial in patients’ adherence to CPAP treatment, as they should not be seen as outsiders, but as integral to the success of the treatment. Partners can have both a positive and negative impact. Couples express the need to support each other and accept responsibility for each other’s wellbeing. Open, frequent, and supportive communication is necessary to facilitate adaptation.
Gibson et al., 2018 [56]	To explore the experience of older people (≥65 years) and their partners, living in the Greater Wellington region, regarding diagnosis and treatment for OSA with CPAP	Qualitative study with focus group. Samples: 25 (16 patients, 15 male and 1 female, and 9 spouses/partners).	Focus group (participants were divided into three groups). Breathing before the focus group.	The partners report the symptoms of OSA to the patient and remind them to perform the therapy. Symptoms are not noticed by most patients until treated with CPAP. The key issue for patients and partners is the noise associated with air leaks and CPAP equipment. Overall feedback from participants is positive about both the effect of CPAP and routine adoption of therapy. All participants report improved quality of life.	The key role of the partner in identifying the symptoms (and variation) of OSA, and in the correct use of devices, emerges.
Jara et al., 2018 [57]	Assessing the association of CPAP with sexual QoL for patients with OSA.	Prospective, observational cohort study. Samples: 182 (115 men and 67 women) Inclusion criteria: age 18–80 years; knowledge of the English language; CPAP therapy; ability to give informed consent; ability and willingness to understand the study protocol. Exclusion criteria: not having a telephone; previously diagnosed sleep disorder; intention to move during the study. CPAP treatment users were defined as those patients who used the CPAP treatment for more than 4 h per night; conversely, non-users were those who used the CPAP treatment for less than 30 min per night. Study length: 1 year.	Patients were met prior to initiation of CPAP treatment, and at 12 month follow-up. CPAP use was assessed via the device’s internal memory card. Tools: QoL change → SNORE-25.	The cohort is divided into 72 CPAP users and 110 non-users. Both groups show an improvement in QoL at 12 months (users: at baseline 1.2; SD = 1.1; and at follow-up 0.5; SD = 0.7). Men show no difference in improvement of QoL at 12 months between CPAP users and non-users (*p* > 0.05), in contrast to women who show a greater improvement of QoL at 12 months in CPAP users (*p* < 0.05).	CPAP users show an improvement in QoL compared to non-users. Among CPAP patients, women reveal a significant association between CPAP and QoL, in contrast to men.
Ward et al., 2018 [58]	Exploring experiences of living with CPAP therapy in accompanied persons with OSA.	Grounded theory study. Sample: 16 (9 men and 7 women) Exclusion criteria: age < 17 years; patients prescribed CPAP for other diseases.	Semi-structured interview conducted by telephone, with me, duration of 52 min. The interviews started with open questions to elicit information about the participant regarding CPAP.	Themes that emerged are: becoming a team that sleeps well, making choices about CPAP, and getting used to CPAP. “Becoming a team” explains how patients using CPAP see the role of their partners. Participants also report what contributions are made by family members and friends. Participants and their partners make changes in lifestyle habits to incorporate CPAP into daily life. The presence of family members is found to be the cause of the identification of the health problem (“for years and years, I didn’t see it as a big problem, because I never counted apneas”); they are also credited with the success of treatment adherence.	The study led to the construction of a theory based on negotiating the positive and negative aspects of CPAP, and the balance of living with it. Partners are an integral part of the CPAP process, and should be included in the pathway from diagnosis to treatment management at home. The active presence of partners promotes understanding of the pathology, purpose, and management of therapy.
Gentina et al., 2019 [59]	Assess spouse/partner involvement and relationship quality on CPAP adherence.	Multicentre prospective study. Samples = 290 (224 males and 66 females) Inclusion criteria: age >18 years; cohabitation > 1 year; newly diagnosed OSA, with no previous experience of CPAP use. Exclusion criteria: patients with neurocognitive disorders; patients with language fluency problems. A total of 72.4% of couples report sleeping in the same room. Length of study: 3 months.	Standardized 1 h educational programme, including 10 min videos on OSA, before starting treatment. At the beginning of treatment, patients assessed their marital relationship → QMI. After 45 days, patients completed a questionnaire to assess their partner’s commitment to CPAP. The following were assessed: pressure to use CPAP; emotional support; cooperation.	Partner involvement has a direct impact on QoL and adherence to therapy (*p* < 0.05). For patients with a high QMI, the relationship between partner involvement and adherence to therapy is statistically significant (*p* < 0.001).	Partner involvement and relationship quality have a significant impact on CPAP adherence and perceived quality of life, particularly in couples with a high QMI.
Khan et al., 2019 [23]	To identify OSA patients’ preferences, partners’ experiences, barriers and facilitators of CPAP adherence, and to assess understanding of the educational content provided, and satisfaction with the multidimensional intervention.	Randomized controlled clinical trial. Samples: 60 (28 treatment group, 32 control group).	Patients and partners participated in an information session on the use of the device. The experimental group underwent a further thirteen training sessions (60–90 min): interactive educational sessions; coaching; practical exercise with the device by a respiratory therapist; semi-structured motivational interview. The control group was observed over time.	Two positive and two negative emotional themes emerge. Sense of relief and desire to live long; fear and frustration. Part of the sample report that they started treatment prompted by their partner. Peer-coaching demonstrates the potential value of an emotionally supportive environment for treatment adherence. Adherence to treatment closely linked to the existence and quality of family ties	Patients’ and partners’ positive experiences with CPAP are enhanced by patient-centered education, and improved adherence to therapy. It is important to address the couple’s fears and concerns in order to optimize therapy.
Luyster et al., 2019 [29]	Assessing the acceptability, feasibility, and preliminary effectiveness of a CES intervention for CPAP adherence.	Observational quantitative pilot study Sample: 30 (patients/partners) Inclusion criteria: age > 18 years; diagnosis of OSA; CPAP treatment; mothers or cohabitation with partner; English reading and writing skills. Exclusion criteria: previous CPAP treatment. Length of the study: 3 months.	Random assignment to three groups: CES, PES, and UC. CPAP adherence assessed by memory device at 1 week, 1 month, and 3 months. Tools: daytime sleepiness → ESS; sleep quality → PSQI; impact of sleepiness on life activities →FOSQ-10.	CES: increased adherence to CPAP (1.4 h from 1 week to 1 month; decrease of 1.6 h from 1 to 3 months). In the three groups, improvements are observed at 3 months for ESS, PSQI, and FOSQ-10. Specifically for the CES group, PSQI: at baseline 8.4 (SD = 2.5), and at 3 months 5.0 (SD = 3.4); EES: at baseline 9.0 (SD = 6.9), and at 3 months 4.2 (SD = 0.8); FOSQ-10: at baseline 16.6 (SD = 2.6), and at 3 months 17.9 (SD = 0.9). The partners also report improved values on all three scales from baseline to 3 months.	An educational and supportive intervention aimed at new CPAP users and their partners is feasible and beneficial. Improvements in sleep quality, daytime sleepiness, and daytime function are evident for both patients and partners in the CES group. In the CES group, there is an increase in CPAP use in the first month, and a decrease at 3 months. The authors recommend further educational sessions. All partners report that the intervention helped them to support the patient in using the CPAP. This involvement may decrease with time.
Adams et al., 2020 [60]	Explore the role of two important interpersonal descriptors (attachment and relationship satisfaction) on treatment initiation and CPAP compliance. The benefit of CPAP treatment on sleep measures and psychological functioning is also examined.	Observational study pre-test post-test. Sample: *N* = 83 (T1) (69 women and 14 men), after 3 months (T2) *N* = 31 Inclusion criteria: age 18–65 years; diagnosis of OSA; living with partner for six months. Exclusion criteria: other sleep disorders; hypnoinductive therapy at the time of assessment; substance abuse; oxygen therapy; recent hospitalization; history of heart failure, chronic pain, chronic obstructive pulmonary disease, psychosis, or bipolar disorder.	Tools: adherence to therapy → ECR; sleep quality → PSQI; daytime sleepiness → ESS; depressive symptoms and anxiety symptoms → PHQ-9 and GAD-7; relationship satisfaction → CSI-16 Adherence to CPAP treatment was recorded on a memory card in the CPAP machines (between T1 and T2).	There is no significant difference in the perception of anxiety between those who started the treatment prompted by their partner or autonomously (*p* = 0.049), and those who decided in agreement with their partner or autonomously (*p* = 0.04). Adherence to CPAP increases at T2, associated with increased use of the device (*p* = 0.02). Between T1 and T2, improved sleep quality and reduced levels of depression (*p* < 0.001) and anxiety (*p* < 0.001). No statistically significant difference between T1 and T2 in CSI (*p* = 0.26)	The use of CPAP makes a significant difference in the treatment of sleep, a reduction in depression and anxiety at 1 month after treatment, but these are not sustained after 3 months of treatment. The study does not provide relevant information with respect to the improvement in the quality of the couple’s relationship 3 months after the start of treatment. The authors suggest including the partner in CPAP education and management.
Baron et al., 2020 [53]	Examining patients’ perceptions of partner support on CPAP adherence.	Quantitative observational study. Sample: 92 patients Inclusion criteria: OSA diagnosis; cohabitation > 1 year; no CPAP use. Exclusion criteria: treatment for OSA other than CPAP; chronic obstructive pulmonary disease; oxygen therapy; congestive heart failure; cardiomyopathy; psychosis; non-native English speaker. Study length: 2 months.	Completion of online questionnaires at 14 and 60 days after CPAP initiation: support for perceived partner autonomy and response to CPAP. Adherence to CPAP was assessed through the device memory. Tools: autonomy support → IOCQ; perceived partner participation → Likert scale.	Average daily increase in CPAP use (*p* < 0.001). A significant improvement in perceived partner support is observed at 14 and 60 days (*p* = 0.046; *p* = 0.001). The association between partner autonomy support and CPAP use is re-evaluated at 1, 14, 30, 46, and 60 days: day 1: *p* = 0.546; day 14: *p* = 0.046; day 30: *p* = 0.001; day 46: *p* < 0.001; day 60: *p* = 0.001.	Positive and/or negative spousal attitudes discriminate against CPAP adherence. Perceived autonomy is observed to be associated with CPAP adherence, with patients reporting higher levels of partner support being more likely to adhere to treatment.

AHI: apnea-hypopnea index; BDI: Beck depression inventory; BMI: body max index; CES: couples-oriented education and support; CPAP: continuous positive airway pressure therapy; CSI: couples’ satisfaction index; DAS: dyadic adjustment scale; DASS: depression anxiety stress scales; ECR: experiences in close relationship; ESS: Epworth sleepiness scale; ESSI: enhancing recovery in coronary heart disease index; FOSQ-10: functional outcomes of sleep questionnaire; FSDS: female sexual distress scale; FSFI: female sexual function index; GAD-7: generalized anxiety disorder-7; IIEF: international index of erectile functional; IOCQ: important other climate questionnaire; LISAT-11: life satisfaction 11; MFSD: manifest female sexual dysfunction; OSA: obstructive sleep apnea; PES: education and support intervention directed only at the patient; PHQ-9: patient health questionnaire-9; PSG: polysomnography; PSQI: Pittsburgh sleep quality index; QMI: quality of marriage index; QoL: quality of life; QRI: quality of relationship inventory; SD: standard deviation; SES: socioeconomic status; and UC: usual ca.

## Data Availability

Data sharing is not applicable to this article.

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
