# Peer review of "Obstructive Sleep Apnea and Adherence to Continuous Positive Airway Pressure (CPAP) Treatment: Let’s Talk about Partners!"

_healthcare, 2022, doi:10.3390/healthcare10050943_

Round 1

Reviewer 1 Report

I have read the article by Rosa et al. with great interest. Adherence to CPAP is a hot topic. The authors have investigated an important factor, the influence of partners on this.

Comments:

  • CPAP is not ventilation. Please, correct.
  • “Although CPAP improves the patient's cardiovascular, metabolic and inflammatory parameters and reduces the risk of cardiovascular morbidity and mortality”. I do not think that it is fully true. Please, revise by softening this statement.
  • Ref 63 is inappropriate as it covers ASV rather than CPAP.
  • Introduction and Discussion. I am curious to see that the overall compliance is so low (i.e. 40%), whilst many other real life studies (i.e. analysing ResMed database or Medicare data) are showing much better figures. Please, comment.
  • Throughout the manuscript it seems that the authors focus on traditional partnership model emphasising supportive wives. Do you have any data on LMBTQ relationships? Are they different? Is there any data. Please, comment briefly in the manuscript.  

Author Response

The authors thank the reviewer for the important review work.

Response to comments follows

  • CPAP is not ventilation. Please, correct.

Thank you for your comment. To avoid misunderstandings, we have removed the word ventilation from the text.

  • “Although CPAP improves the patient's cardiovascular, metabolic and inflammatory parameters and reduces the risk of cardiovascular morbidity and mortality”. I do not think that it is fully true. Please, revise by softening this statement.

Thank you for your comment, we have rewritten the sentence

  • Ref 63 is inappropriate as it covers ASV rather than CPAP.

Thank you for your comment, we have removed ref 63

  • Introduction and Discussion. I am curious to see that the overall compliance is so low (i.e. 40%), whilst many other real life studies (i.e. analysing ResMed database or Medicare data) are showing much better figures. Please, comment.

Thank you for the interesting comment. In the introduction we have supplemented the data found in the literature with those suggested by the reviewer.

  • Throughout the manuscript it seems that the authors focus on traditional partnership model emphasising supportive wives. Do you have any data on LMBTQ relationships? Are they different? Is there any data. Please, comment briefly in the manuscript.

We did not find any data on LGBTQ couples, so we decided to include the comment in the limits and discussion of the review.

.

Reviewer 2 Report

Although it is clear that a great deal of work went into this paper, I think the language and organization could more clearly delineate the role of the bed partner in improving OSA outcomes. I commend the authors on their tremendous efforts, but would like to raise the following concerns:

Major:

1) Extensive English revision is needed. Often times, the word choices and punctuation make the paragraphs hard to follow and greatly detract from the readability. 

2) A succinct summary of the parameters improved through partner engagement (perhaps as a table) should be included. For example, encouraging partner engagement is cited as a means of improving CPAP use, but what other domains of quality of life are improved? For that matter, what didn't work? These things are alluded to within the text, but are strung together in a manner that is difficult to follow. Putting these key take-homes in one place will improve the readability of the manuscript.

3) Overall, I would recommend restructuring the manuscript with a focus on positive partner engagement, as this seems to be the greatest focus of the text. As things stand, there are a variety of modalities/styles of engagement noted with outcomes listed variably. 

Author Response

The authors thank the reviewer for the important review work.

  • Response to comments follows Extensive English revision is needed. Often times, the word choices and punctuation make the paragraphs hard to follow and greatly detract from the readability. 

Thank you for your comment, we have sent the paper for language review

  • A succinct summary of the parameters improved through partner engagement (perhaps as a table) should be included. For example, encouraging partner engagement is cited as a means of improving CPAP use, but what other domains of quality of life are improved? For that matter, what didn't work? These things are alluded to within the text, but are strung together in a manner that is difficult to follow. Putting these key take-homes in one place will improve the readability of the manuscript.

Thank you for your comment. Unfortunately, the results of the analysed papers are very heterogeneous and we were not able to generate a table.

  • Overall, I would recommend restructuring the manuscript with a focus on positive partner engagement, as this seems to be the greatest focus of the text. As things stand, there are a variety of modalities/styles of engagement noted with outcomes listed variably. 

Thank you for your comment, we have split the last paragraph "barriers and facilitators" to focus on the positive partner engagement.
